# Hidden units for tabular data representing intervals

## Abstract

Tree-based boosting remains a strong baseline for tabular data, partly because standard neural units impose overly smooth inductive biases. We revisit exponential-centered units (ExU) through the lens of Lipschitz bounds and introduce Double-centered Units (DcU), which parameterize soft intervals via learnable left/right centers and preserve informative gradients outside the interval. Building on DcU, we propose the Soft Interval Neural Network (SINN)—an encoder-MLP architecture with max pooling and interval sparsity regularization. Across 15 public datasets, SINN delivers competitive or superior accuracy to XGBoost on classification, while performance on regression is more mixed; we hypothesize that this gap reflects the implicit bias of neural networks. We further examine common generalization proxies—spectral/Lipschitz bounds, Hessian-based flatness, and dropout-based ensembling—and find that smoothness-oriented regularization is not consistently predictive of tabular performance. These results suggest that non-affine, interval-like representations provide a useful inductive bias for tabular classification, and motivate theoretical analyses beyond affine assumptions.

## 1 Introduction

Neural networks have contributed to remarkable advances in various domains of artificial intelligence such as He et al. (2016); Devlin et al. (2019). However for tabular data, tree-based boosting models (hereafter, tree models), such as XGBoost(Chen & Guestrin, 2016) and Cat-Boost(Prokhorenkova et al., 2018) are widely used in real-world applications(Kaggle, 2020). This is because tree models are fast to train, interpretable and perform better overall(Shwartz-Ziv & Armon, 2022; Borisov et al., 2022). The dominance of tree models is explained by Grinsztajn et al. (2022), in which the authors argue that neural networks are too smooth to learn tabular representations, implying that tabular data possess distinct properties compared to images or natural language.

In this paper, we propose novel hidden units and a corresponding supervised neural network architecture for tabular data. The hidden units significantly increase their Lipschitz bound(or Lipschitz constant if we abuse the term, since it is not the tightest upper bound), by expanding a vector(scalar) into a matrix(vector). We call them Double-centered Units (DcU), as the improvement over Exp-centered Units(Agarwal et al., 2021) is the one extra center parameter, making them resemble intervals. We build a simple architecture called SINN, using DcU, and conduct extensive supervised learning experiments showing that SINN achieves comparable performance to XGBoost.

Furthermore, we study the property of tabular data in view of generalization. Generalization refers to the expected loss under a true distribution, which is approximated using a held-out test set since the true distribution is inaccessible. Its properties have been studied in various perspectives: Model-complexity, parameter space smoothness and averaging of models, all suggesting that the smoothness is a key measurement for the generalization of models- whether in terms of output changes, parameter space, or the contribution of individual model components. On the contrary, we hypothesize that we hypothesize that excessive smoothness can hinder tabular performance, because DcU rely on their rapid changes of output and sudden changes of gradients. We study how these measurements behave for SINN.

## 2 RELATED WORKS

In this section we introduce 3 views on generalization of models and corresponding methods for neural networks. The measurements for generalization are summarized in Table 1.

Table 1: Generalization measurements described in section 2.2

| Item | Related | Meaning |
|---|---|---|
| Singular value | Lipschitz constant(2.1) | The sum of largest singular values of each layers' weight |
| Hessian | Flat minimum(2.2) | The largest eigenvalue of loss hessian calculated with training data |
| Dropout | Averaging(2.3) | Average dropout rate, selected by Hyperopt |

### 2.1 LIPSCHITZ CONSTANT

First, we briefly review generalization from the PAC-learning perspective (Chapter3 of Mohri et al., 2018): Let $\mathcal{X}$ and $\mathcal{Y}$ denote the input and label spaces, and let $\mathcal{D}$ be a distribution over $\mathcal{X} \times \mathcal{Y}$. Given an i.i.d sample $\mathcal{S}$ of size m drawn from $\mathcal{D}$, a learning algorithm $\mathcal{A}$ returns a hypothesis $h : \mathcal{X} \to \mathcal{Y} \in \mathcal{H}$(we write $h_{\mathcal{S}}$ to emphasize its dependence on $\mathcal{S}$). Expected error of $h$ over $\mathcal{D}$ and Empirical error of $h$ over $\mathcal{S}$ are denoted as $E_{(X,Y)\sim\mathcal{D}}\left[L(Y, h(X))\right]$ and $\hat{R}_{\mathcal{S}}(h) = \frac{1}{m}\sum_{(x,y)\in\mathcal{S}} L(y, h(x))$. Given that $L$ is classification error(1-Accuracy) and $\mathcal{Y} = \{-1, 1\}$, the two quantities are related as below:

$$R(h) \leq \hat{R}_{\mathcal{S}}(h) + \text{complexity}(\mathcal{H}) + \mathcal{O}(1/m) \tag{1}$$

The complexity($\mathcal{H}$) should increase as there exists a $h \in \mathcal{H}$ that can fit any distribution over $\mathcal{X}$ and $\mathcal{Y}$. The extreme and intuitive case is a full random data; $\mathcal{X}$ and $\mathcal{Y}$ are sampled from random distributions in an i.i.d fashion. The popular choice for the complexity is VC dimension(Vapnik, 2013). However when $\mathcal{H}$ is a set of linear models in Reproducing Kernel Hilbert Space(RKHS), the interesting result can be derived, once $\mathcal{X}$ is scaled to 1:

$$R(h) \leq \hat{R_{\mathcal{S},\rho}}(h) + \frac{\sqrt{2}\Lambda}{\rho} + \mathcal{O}(1/m) \tag{2}$$

,where $\Lambda$ is the upper bound of the hilbert norm a weight vector(normal vector) of $h$ and the empirical margin loss $\hat{R}_{\mathcal{S},\rho}(h) = \frac{1}{m}\sum_{i=1}^{m} \Phi_\rho(y_i h(x_i))$ and $\Phi_\rho(z) = min(1, max(0, 1 - \frac{z}{\rho}))$(See Figure 1-(a)).

In the world of neural networks, Inequation 1 does not seem to work. Zhang et al. (2021) showed that in an image classification, neural networks able to fit full random data also generalize quite well. Their fits on full random data are not hindered even after regularization methods(frobenius norm, dropout etc) are applied while their generalization also rarely benefit from the methods. On the other hand, Bartlett et al. (2017) suggests a Spectrally-normalized margin bound(Theorem 1.1 of Bartlett et al. (2017)), similar to Inequation 2. The bound is derived using the covering number for the sequence of affine transformations and relating it with the Rademacher complexity. The bound says that if a neural network is well designed so that it separates data well and does not changes its output rapidly with respect to its input, it generalizes well. Surprisingly, Bietti & Mairal (2019) showed that Convolutional Neural Networks(CNN) are in RKHS and their norm in the hilbert space is upper bounded by the product of spectral norms of their weight matrices. This justifies the similarity between Inequation 2 and the Spectrally-normalized margin bound.

There have been studies on regularizing the spectral norm(Cisse et al., 2017; Sedghi et al., 2018; Yoshida & Miyato, 2017). The Spectral norm of a matrix $\boldsymbol{W}$ is the largest 2-norm value of matrix-vector products scaled by the norm of vectors:

$$\|\boldsymbol{W}\|_\sigma = \max_{\boldsymbol{x}\neq 0} \frac{\|\boldsymbol{W}\boldsymbol{x}\|_2}{\|\boldsymbol{x}\|_2} \tag{3}$$

, and coincides with $\boldsymbol{W}$'s largest singular values. Yoshida & Miyato (2017) approximates it using a power iteration method to calculate the largest singular value of weight matrices of a neural network.

### 2.2 FLATNESS OF MINIMUM

Another measurement of the complexity of a model $h$ is the smoothness of its parameter space over loss function. The relation with the generalization is illustrated in Figure 1-(b) - in a smooth parameter space, loss function value does not change much when a distribution shifts in test data(Hochreiter

& Schmidhuber, 1997). The rationale behind is Minimum Description Length(MDL) theory, which states the sharp minimum should require more bits to be specified, making it more complex and thus prone to poor generalization. Hochreiter & Schmidhuber (1997) suggested an Flat Minimum Search(FMS) algorithm to encourage networks to converge to a flat point. Flatness is characterized by a low curvature thus expressed as small eigenvalues of a Hessian matrix.

It has been widely observed that neural networks trained with small batch sized stochastic gradient (SB) methods generalize better than those trained with large batch (LB) sized stochastic gradient methods. From the fact that LB and SB methods exhibit similar train errors, Keskar et al. (2016) hypothesized and showed that SB converges to flat minimum. In addition they also showed that SB methods explore a wide range of parameter space while LB method settle settle into minima closer to an initial poin. We adopt SB methods to flatten our loss function, since FMS algorithm was devised only for regression. Moreover, batch size is one of SINN's hyperparameter, thus it is important to investigate both whether and why it plays a critical role.

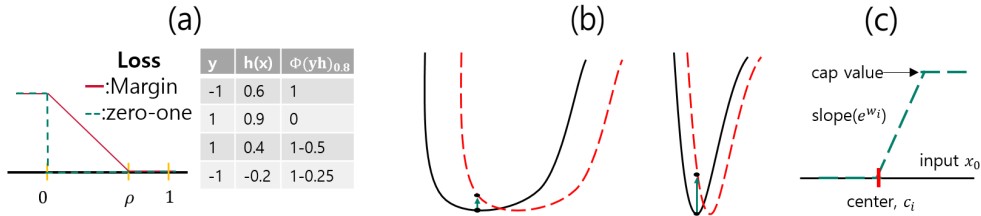

Figure 1: (a) Illustration of margin loss with $\rho = 0.8$. Note that the margin loss is always greater than zero-one loss(classification error) and large $\rho$ requires more confidence not to be an error thus tends to increase the loss value. (b) Black line is loss value of training data and red dot-line is loss value of test data. If minima is smooth, a shift of distribution does not change the loss value much. (c) A function value of a single unit of ExU

## 2.3 Averaging

Instead of a single $\mathcal{S}$, one can consider the expectation of $R(h_\mathcal{S})$ over $\mathcal{S} \sim \mathcal{D}^m$, which decomposes into Bias$^2$ + Variance(Chapter 3 of Bishop & Nasrabadi (2006)); Bias is the difference between the true function and the average learned functions and Variance is the variability among the learned functions. A flexible algorithm generates models having low bias but fluctuating when small changes in training data occur. Thus Bias and Variance is in trade-off, suggesting that the algorithm should adjust its complexity. In fact, this mirrors the conclusion of Section 2.1; both of which favor a simple $H$. Theoretically, Bias and Variance can be reduced simultaneously by averaging unrelated and low-biased functions. However, this is infeasible in practice because we typically only have a single training dataset $\mathcal{S}$ and cannot resample from a true distribution. Bagging is a method to avoid it and tree models employ techniques like subsampling(Breiman, 2001; Friedman et al., 2003). Note that unlike above two perspective, averaging does not rely on a norm of input vectors or of a parameter space. Unlike tree models, bruteforce averaging of neural networks seems impossible, but Srivastava et al. (2014) devised a creative method to ensemble networks, called Dropout. It disconnects all edges from and to a unit with a fixed probability($p$), producing thinned networks at a train stage. At inference stage, no edges are dropped. Then the final network becomes an ensemble of thinned networks. Specifically, it scales down by $p$ so that the expected values of units remain same at inference.

## 3 Suggested Architecture

In this Section, we unveil the reason behind the success of ExU and DcU, in terms of their Lipschitz constant in Appendix B. In short, they exhibit a distinct property over MLP, having large Lipschitz bounds as demonstrated in Equations 4 and 5. Although they share a core non-affine transformation, we find the training of ExU is somewhat hindered, demonstrated by its small gradients. DcU is designed to activate their gradients. We then propose an encoder and neural network architecture using DcU for supervised learning.

### 3.1 EXPONENTIAL-CENTERED UNITS

ExU are defined for a scalar $x \in R$ with two parameters $\boldsymbol{w}, \boldsymbol{c} \in R^h$(weight, center) $x \mapsto \sigma(e^{\boldsymbol{w}} \odot (x - \boldsymbol{c})) \in R^h$, where $\sigma\cdot$ is an activation function, $x - c = x\mathbf{1} - \boldsymbol{c}$, $\mathbf{1}$ is one vector in $R^h$, $e^{\cdot}$ is an element-wise exponential and $\odot$ denotes Hadamard (element-wise) product. Agarwal et al. (2021) used clipped ReLU for $\sigma$.

The authors mentioned that it enables a sharp change of output with a tiny input change to convert $w$ into exponential term. Instead, we believe that linear transformation is what distinguishes ExU from standard hidden units(ShU, $\boldsymbol{x} \mapsto \sigma(\boldsymbol{W}\boldsymbol{x} + \boldsymbol{c})$ for $\boldsymbol{x} \in R^d, \boldsymbol{W} \in R^{h \times d}$). Both units can, in principle, generate same-valued outputs since there exist parameter sets that make two vectors' values the same. However, as shown in Appendix B, their mathematical properties with respect to the Lipschtiz bounds differ substantially – ExU are more sensitive to input changes, while ShU must be deep in order to be sensitive. To support our hypothesis, we trained three different units on a Bernoulli data(Appendix A), following the original paper; ExU, noExU (ExU without exponential weight) and ShU. See results in Table 2: Shallow ShU struggles to learn certain hyperplanes and even deep ShU cannot fit well. ExU and noExU fit the data well only with large initial weights. The experiment indicates that the Lipschtiz bound for a Multilayer perceptron (MLP) is not tight and ExU has a unique representation.

Table 2: Training results on Bernoulli data with different Hidden units. Units are denoted as follows: ShU-$h$-$L$ refers to $h$ hidden units and $L$ layers. ExU-$\mu$ and noExU-$\mu$ refer to $\mu$-mean truncated normal distribution with standard deviation of $\mu/10$ for weight initialization. DcU-$s$ refers to zero-mean and $s$-standard deviation as their weights initialization. Note that the direct calculation of sample probability and XGBoost with 10,000 weak learners have loss value of 0.58108 and 0.58114 respectively. Lipschitz denote log value of Lipschitz bound in Equations 4 and 5

| Units | ShU-50-10 | ShU-10000-1 | ExU-1 | ExU-4 | noExU-30 | DcU-0.1 | DcU-0.5 | DcU-1.0 |
|---|---|---|---|---|---|---|---|---|
| Loss | 0.5858 | 0.6662 | 0.6270 | 0.5815 | 0.5819 | 0.5810 | 0.5810 | 0.5812 |
| #Param | 23101 | 30001 | 3001 | 3001 | 3001 | 501 | 501 | 501 |
| Lipschitz | 6.154 | 1.138 | 4.781 | 5.522 | 5.415 | 5.366 | 5.480 | 5.600 |

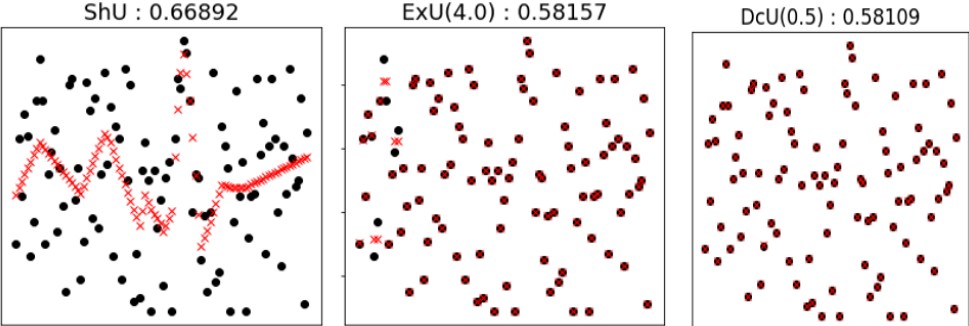

Figure 2: Results of fitting on a Bernoulli data, Lipschitz refers to the Lipschitz bound calculated as Appendix B

From the above experiment, we showed that large weights are important for ExU's learnability. To explain why, we provide what a single unit of ExU expresses(Figure 1-(c)). It can represent an interval, as the unit represents whether the input lies on left or right to the interval and has increasing values from 0 to 1 between the interval. Note that the interval is determined by a center and a weight(or slope). This process is similar to what decision tree split learns. Therefore a large weight keeps unit value to stay in a binary(0 or 1) area. Figure 3-(a) shows that the units of ExU with large initializations are binarized. Our concern is that when the slope is too sharp, gradients may vanish across almost every point halting learning on ExU parameters: in Figure 3-(a), gradients of $w, c$ drop rapidly. Moreover ExU require 1000 units to fit the Bernoulli data, implying that their units are of little use.

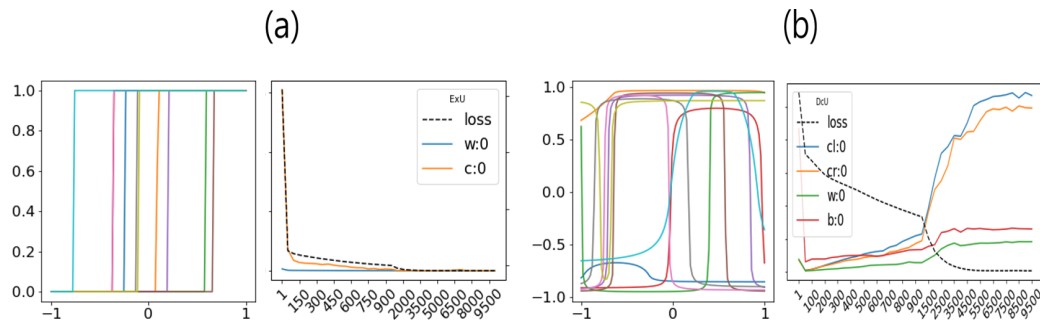

Figure 3: (a) The figure represents (left) 10 random ExU units and (Right) Training loss and Average value of Gradient's norm of ExU. (b) The figure represents the same as (a) but for DcU

## 3.2 DOUBLE-CENTERED HIDDEN UNITS

To overcome this problem, we develop Double-centered Units (DcU). The key idea is that gradients should not vanish outside the interval. Given a true interval (a,b), the parameters should be adjusted until they represent the interval – the opposite of what occurs in ExU. Here, we present the formulation of DcU for a scalar input $x$: With 4 parameters $\boldsymbol{w}, \boldsymbol{c}^l, \boldsymbol{c}^r, \boldsymbol{b} \in R^h$ and a hyperparameter $\beta \in R^+$:

$$x \mapsto \sigma_\beta(\boldsymbol{w} \odot (\sigma_\beta(x - \boldsymbol{c}^l) + \sigma_\beta(\boldsymbol{c}^r - x)) + \boldsymbol{b})$$

, where $\sigma_\beta(\cdot) = \sigma(\beta * \cdot)$ and $\sigma(\cdot)$ is a softsign. A unit acts like a soft interval, whose left and right edges are denoted by $\boldsymbol{c}^r$ and $\boldsymbol{c}^l$. The rationale is illustrated in Figure 4. Only with softsign and an extra activation, a unit maintains gradients outside the true interval (-0.3,0.3). Other symmetric activation functions cannot maintain gradients or have gradients that are too small. Although ReLU maintains gradients outside the interval, it is not symmetric, which disables the Sparsity regularization we describe later.

The learnability of DcU is explored with the same Bernoulli data(Figure 2). DcU fit the data robustly to initialization and do not suffer from gradient vanishing, as their units are not binarized(Figure 3-(b)). In addition, DcU require fewer dimensions($h = 100$, 1/10 of ExU) and achieve an almost perfect fit.

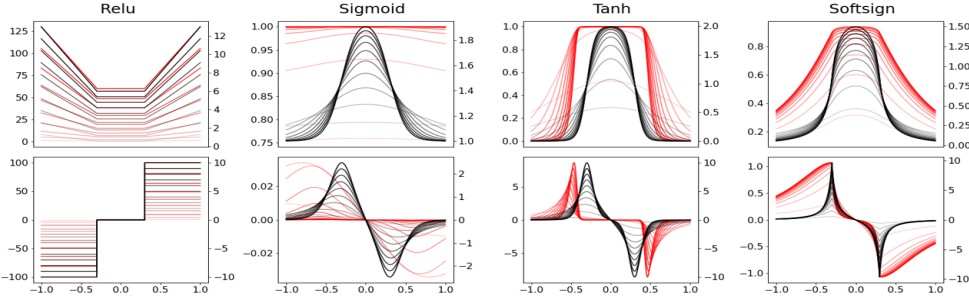

Figure 4: DcU illustration by activation functions, $\beta$ and extra activation. Red line(Left y-axis) and Black line(Right y-axis) are ones with and without extra activation function. The color gets light as $\beta$ gets small(1 to 10).

## 3.3 SOFT INTERVAL NEURAL NETWORKS

Extension to a multivariate case can be done by stacking scalar DcUs to form a matrix. To ensure the versatility of this matrix, it must be mapped into a vector. The entire layer, from a d-dimensional input vector to a h-dimensional embedding vector, is called a Soft Interval Layer (SIL). The Lipschitz bound of SIL is explained in Appendix B. It operates as follow: For an input vector $\boldsymbol{x} \in R^d$, it has 5 parameters($\boldsymbol{W}, \boldsymbol{C}^l, \boldsymbol{C}^r, \boldsymbol{B} \in R^{d \times h}$ and $\boldsymbol{M} \in R^{h \times h}$):

$$\boldsymbol{A} = \sigma_\beta(\boldsymbol{W} \odot (\sigma_\beta(\boldsymbol{x} - \boldsymbol{C}^l) + \sigma_\beta(\boldsymbol{C}^r - \boldsymbol{x})) + \boldsymbol{B}) = \begin{pmatrix} \boldsymbol{A}_{1,:}^T \\ ... \\ \boldsymbol{A}_{d,:}^T \end{pmatrix} \in R^{d \times h}$$

$$\boldsymbol{a} = R_{max}(\boldsymbol{AM}) = R_{max}(\begin{pmatrix} \boldsymbol{A}_{1,:}^T \boldsymbol{M}_{:,1} & ... & \boldsymbol{A}_{1,:}^T \boldsymbol{M}_{:,h} \\ & ... & \\ \boldsymbol{A}_{d,:}^T \boldsymbol{M}_{:,1} & ... & \boldsymbol{A}_{d,:}^T \boldsymbol{M}_{:,h} \end{pmatrix}) \in R^h$$

,where for a matrix $\boldsymbol{X}$, $\boldsymbol{X}_{:,j}, \boldsymbol{X}_{i,:}$ represents a $j$th column vector and $j$th row of any matrix $\boldsymbol{X}$. Thus $A_{i,:}$ is an output of a scalar($x_i$) DcU. $R_{max}(\boldsymbol{X})$ is the operation that reduces each $\boldsymbol{X}_{:,j}$ into its max values. $\boldsymbol{M}$ re-maps elements of each $\boldsymbol{A}_{i,:}$ into another $h$ dimension, unlike NAM(Agarwal et al., 2021) which re-maps outputs into a scalar, enabling feature selection across each $h$ dimension. Dropout is applied to $\boldsymbol{A}$.

Since SIL is an encoder, the subsequent layer can then take it an input. We attach a Multilayer perceptron (MLP) and a link function (Link) in order and call the entire architecture Soft Interval Neural Network (SINN, Figure 5-(a)). Other details on SINN are provided in Appendix C.

We encourage sparsity on $\boldsymbol{A}$, as the number of units is same across all features, which contradicts

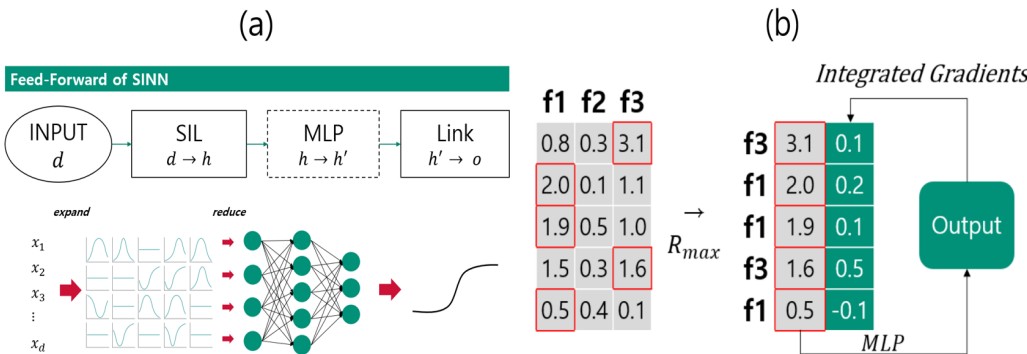

Figure 5: (a) Visualization of SINN (b) Explainability module of SINN. Gray cells represent unit values. Red boundary means the cell is selected via $R_{max}$. Green cells represent attribution values calculated by Integrated Gradients. For this case, attribution values of each features are **f1**=0.1,**f2**=0.0,**f3**=0.6

the fact that feature importance varies. To achieve the goal, we assign a Interval Regularization: L1 and L2 norms on $\boldsymbol{C}_l - \boldsymbol{C}_r$. Note that only with a symmetric activation $\sigma(x) = -\sigma(-x)$, $\boldsymbol{c}_l = \boldsymbol{c}_r$ leads to a flat line.

Thanks to its simplicity, SINN can provide feature contributions to outputs in two straightforward step. First, we apply the Integrated Gradients(Sundararajan et al., 2017) over $a$ and calculate the attribution values for each units. Second, we aggregate the attribution values of units according to the features they represent. This is possible since we use a $R_{max}$ so that $i$-th unit of $a$ originates from a single feature. Figure 5-(b) illustrates this process. Summing all values of each cases in training data, in an absolute term, would lead to (global) feature importance. As we will demonstrate in Section 4, the result is quite similar to that of XGBoost.

As we extend DcU to a multidimensional case, we train SINN on random data as in Zhang et al. (2021) but lying on fewer dimensional spaces. Compared to a tree model(XGBoost), SINN and MLP exhibit much tight fits on random data(Table 4). However, as dimensionality decreases, the learnability of MLP deteriorates, whereas SINN maintains its performance. This can be explained using the implicit bias of neural networks. Soudry et al. (2018) proved that neural networks on cross entropy loss optimized via Gradient Descent maximize L2 margin, and MLP cannot enlarge its margin in small dimensional spaces, due to its smooth affine transformation. In contrast, SIL as an encoder performs non-smooth operations, allowing SINN to expand its margin even in small-dimensional space. Note that MLP also has a first few MLP as its encoder.

Robustness with respect to dimensionality is arguably desirable in tabular data, which contain many subgroups, identified by only a few features. For instance, a wealthy group, identified only by income and asset does not go bankruptcy easily, while normal group should be inspected more; transaction patterns and demographic feature etc. We hypothesize that the non-affine expansion provided by SIL enables this property, giving SINN an advantage in tabular data learning.

## 4 EXPERIMENTS

We train and evaluate SINN on a variety of supervised learning datasets. Datasets are described in Table 4. Results are compared to XGBoost and SAINT(Somepalli et al., 2021), which is reported to be a generally-best neural network model in Borisov et al. (2022). The procedure of experiments are reported in Appendix D.

Table 3: Description of datasets and performance: RMSE means Root Mean Squared Error. Performance is measured on test set. California is scaled down by 10,000.

| Data | Metric | Size | Feature | Split | Performance | | |
|------|--------|------|---------|-------|-------------|------|-------|
| | | | | | SINN | XGB | SAINT |
| Adult | Accuracy | 48,842 | 14 | 54:13:33 | 86.21(0.14) | 87.31(0.05) | 86.15(0.21) |
| Blastchar | Accuracy | 7,043 | 19 | 70:10:20 | 80.68(1.24) | 80.37(1.40) | 78.82(0.64) |
| Churn | Accuracy | 50,000 | 230 | 64:16:20 | 92.65(0.04) | 92.70(0.03) | 91.73(0.04) |
| Census | Accuracy | 299,285 | 40 | 54:13:33 | 95.62(0.03) | 95.84(0.01) | 94.69(0.04) |
| CoverType | Accuracy | 581,012 | 54 | 60:20:20 | 97.34(0.06) | 96.98(0.05) | 96.88(0.67) |
| Eye | Accuracy | 10,936 | 27 | 70:10:20 | 77.80(1.50) | 76.08(0.73) | 73.84(0.64) |
| Gas | Accuracy | 9,873 | 129 | 70:10:20 | 99.60(0.05) | 99.35(0.16) | 99.29(0.14) |
| Gesture | Accuracy | 13,910 | 32 | 70:10:20 | 71.16(0.83) | 69.97(1.66) | 59.25(0.57) |
| Heloc | Accuracy | 10,459 | 23 | 70:10:20 | 73.12(0.70) | 73.04(0.91) | 70.56(1.38) |
| Higgs | Accuracy | 940,160 | 24 | 64:16:20 | 75.91(0.06) | 74.81(0.05) | 75.52(0.07) |
| Shrutime | Accuracy | 10,000 | 10 | 70:10:20 | 86.24(0.47) | 85.93(0.67) | 77.42(2.20) |
| California | RMSE | 20,640 | 9 | 64:16:20 | 4.666(0.059) | 4.537(0.090) | 4.830(0.130) |
| Rossmann | RMSE | 610,235 | 29 | 45:16:39 | 480.97(3.60) | 505.75(7.42) | 514.57(5.69) |
| Year | RMSE | 515,345 | 90 | 72:18:10 | 8.859(0.038) | 8.818(0.031) | 8.975(0.033) |
| Sarcos | RMSE | 48,933 | 21 | 72:18:10 | 1.406(0.062) | 1.135(0.089) | 2.593(0.046) |

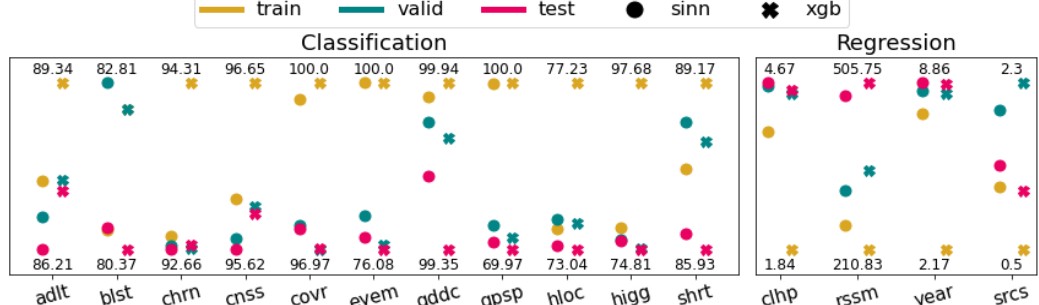

Figure 6: Visualization of performances for training, validation and test data. The datasets are sorted in the same order as Table 3. We mark maximum and minimum values of performances

### 4.1 PERFORMANCE AND REPRESENTATION

Table 3 shows the means and standard deviations of 5 test performances over different seeds, and Figure 6 shows normalized values of training, validation and test performances for each datasets.
In classification tasks, SINN wins overall and if losing by small margin except Adult. Adult contains a large scaled feature with a range of 1.4e+6; after min-max scaling, a DCU unit has difficulty distinguishing values in small-ranged areas (e.g. between 1e+2 and 1e+3). We refer to such features as multiscale feature's , as they contain multiple groups with varying scales (e.g. for income – low, middle, and high-income groups). To validate our hypothesis, we discretize the feature into 100 bins by its quantile. After this transformation, SINN and XGBoost achieve similar performances (86.19 vs. 86.30). Across the other data sets, the performance gap between training and test(or validation) is much smaller for SINN compared to that of XGBoost. In terms of the generalization bound(Equation 2), the difference between training and test performance is narrow when the large margin is guaranteed; as mentioned Section 3.3, neural networks maximize L2 margin while boosting does not. This is verified in Figure 7-(a). In test data, SINN has large absolute margins(mean, median or minimum values), considering the magnitude it has fitted to training data. The difference is evident in Eye, Gas and Gesture. In addition, thanks to L2 margin maximization property, feature

importance is relatively flat compared to that of XGBoost(Figure 7-(b)). Note that although SIL is not linear and not theoretically guaranteed to maximize the margin, SINN has maximized its margin.

Unlike classification, SINN has difficulties in training regression data. SINN matches XGBoost

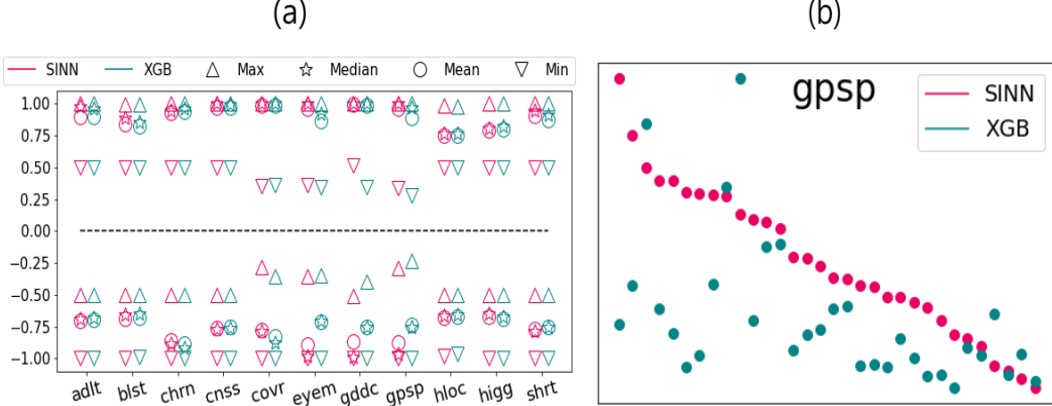

Figure 7: (a) Margin of predictions of SINN and XGBoost. Margin is an its largest output(softmax) but multiplied -1 when it is incorrect. That is, large absolute margin indicates confidence in its prediction. (Right) Normalized Feature importance of Gesture.

only in the datasets where test set is reserved for the prediction to be meaningful(Rossmann and Year, explained in Appendix E). Even without early stopping, we find that SINN cannot reduce its train loss as much as XGBoost does, which means SINN fails to capture true representations. This can again be explained by the implicit bias of neural networks: Being trained on mean squared loss, a neural network, under some assumptions, is known to reduce second derivative with respect to inputs like a cubic spline(Jin & Montúfar, 2023). Although DcU aim to be sensitive to input changes, they fail to overcome this implicit bias. This observation is supported by training regression random data(Table 4); DcU and MLP cannot outperform XGBoost whether targets are scaled to [0,1] but are binary($\{0,1\}$, same as random classification data except loss function), in terms of mean squared loss.

Table 4: Loss for random data of size 10000. Random features and random targets are generated. SINN and MLP trained over 1000 epoch and we retrieve ones with least loss. XGBoost are trained with 20 depths and 10000 weak learners. 'B.' refers to binary

| d | B.Classification | | | Regression | | | B.Regression | | |
|---|---|---|---|---|---|---|---|---|---|
| | SINN | MLP | XGB | SINN | MLP | XGB | SINN | MLP | XGB |
| 5 | 3.2e-10 | 3.0e-03 | 1.3E-03 | 4.3e-04 | 1.6e-03 | 2.9e-06 | 1.0e-03 | 2.5e-03 | 3.0e-06 |
| 10 | 0 | 9.2e-09 | 8.0e-04 | 2.2e-04 | 1.8e-04 | 6.8e-07 | 1.0e-03 | 4.4e-04 | 1.3e-06 |
| 30 | 0 | 0 | 5.1e-04 | 5.9e-05 | 1.3e-05 | 4.9e-07 | 2.2e-04 | 3.7e-05 | 3.3e-07 |

## 4.2 GENERALIZATION MEASUREMENTS

We delve into how generalization measurements described in Table 1 are realized in SINN and their effects on performance in some classification datasets. Let us call the model whose hyperparameters are selected with Hyperopt the best model. For SV and Hess, we apply the corresponding generalization methods to each best models and examine the ratio of measurements, after/(after+before), across the best models(Figure 8-(a)). For dropout, we report the average, minimum and maximum values of dropout rates for SINN and MLP($p^1$ and $p^2$) selected in best models(Figure 8-(b)).

**Singular values** We apply Spectral norm regularization to weights of MLP so that the expected result and theoretical base are obvious, since MLP matches the assumption made in the Spectrally-normalized margin bound(once $a$ is an input to MLP). The regularization reduces singular values, but its impact on performacne is less obvious. This implies that smoothness over even encoded inputs $a$ is not beneficial, just as it is not on the original inputs. Note that this holds true even

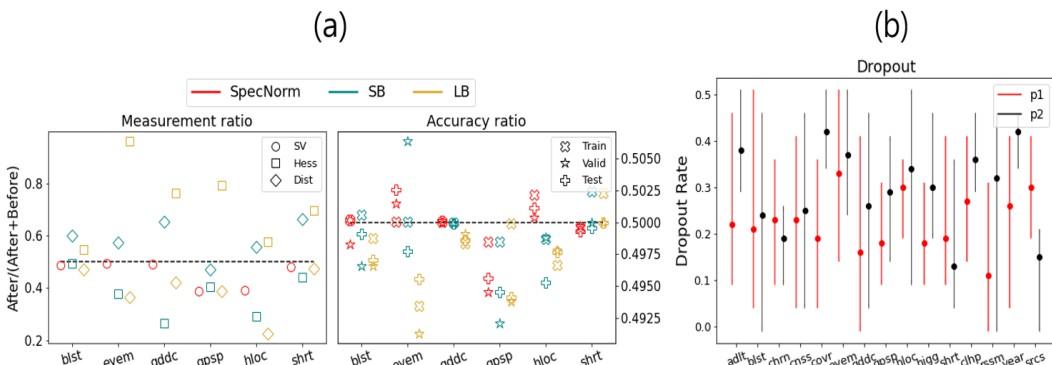

Figure 8: (a) Measurements in Table 1 plus Distance(Norm of trained parameter-initial parameter) are plotted, as ratio 'after/(after+before)'. Therefore 0.5(dotted horizontal line) denotes that a measurement is same after applying regularization methods (b) Min, Max and mean of 5 dropout values selected by Hyperopt

for datasets with all-continuous features(Gas and Gesture). We can conclude even after SIL, the large Lipschitz bound is beneficial for SINN to learn representation. In fact, although we do not provide results here, applying weight decay such as frobenius norm, AdamW(Loshchilov & Hutter, 2017), on weights of MLP or $M$ leads to a substantial drops in both training and test performance, despite being common tools for weight regularization on networks. This suggests that traditional assumptions about weight regularization may not be appropriate for tabular data.

**Hessian** SB methods uses a batch size of 64 and LB methods uses 50% of the training data. The measurement changes as we expect: SB results in the small largest eigenvalue of hessian, calculated via fast multiplication(Pearlmutter, 1994) and move further from the original parameters(Distance). However this does not guarantees better performance. The result seems reasonable, since the loss value is expected to change abruptly as $C^r$ and $C^l$ changes. We should emphasize that unlike image cases, some models are not trained adequately with large batch size. Therefore we conclude batch size is an important hyperparameter, controlling the explorative range of parameter space.

**Dropout** In most datasets, Dropout serves as an essential tool for regularization. This may be because dropout does not depend on metrics, unlike the above two methods. However, under other hyperparameters being fixed, even moderate dropout rates severely hinder training(0.2 or above) – an effect not commonly observed in image domains.

## 5 DISCUSSION

We proposed novel hidden units tailored for tabular data and an architecture utilizing these units. In classification tasks, our approach achieved better results than tree models when margin maximization was effective. Furthermore, we examined how regularization methods affect SINN and provided some insights specific to tabular data. We conclude that smoothness over inputs, including intermediate network outputs, and flat minima are not reliable indicators of generalization, as the underlying smoothness assumptions are not relevant in the tabular. Nevertheless, SINN generalizes well in classification. This suggests that its the generalization properties may be unique and warrant further study. We hypothesize that non-affine transformation are a key factor, since most existing theorems assume affine transformations in neural network operations. In addition, researchers should put much concern on a semi-supervised or a self-supervised learning of tabular data. Since they help supervised learning by introducing a new metric in parameter space(Raina et al., 2007; Jaakkola & Haussler, 1998). However, rationale for their success should be valid only once the smoothness assumption works. This assumption does not make sense in tabular data as we have shown.

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

## A  BERNOULLI DATA

The simulation data is generated in the following order : 1) Generate 100 indices $i \in \mathcal{I} = \{-1, -0.979, ..., 1\}$ with equal width, 2) for each $i$ choose a true probability of event($p_i$) from uniform distribution from 0.1 to 0.9, 3) do a 100 sequence Bernoulli process $y_{(i,1)}, ..., y_{(i,100)}, i \in \mathcal{I}$ and collect them as a data: $\{(i, y_{(i,j)}) | i \in \mathcal{I}, j = 1, ..., 100\}$.

## B  LIPSCHITZ BOUND OF SIL

Here, we show that SIL has distinct nature over a typical Multilayer perceptron (MLP) in light of its Lipschitz bound. The difference mainly results from the expansion operation and the Hadamard product can sharply increase the Lipschitz constant. We use some lemma(Lemma A.1 and Lemma A.2 both in Bartlett et al. (2017)) and properties of matrix norm. Notations are same as explained in the previous sections, unless specified.

A function $f$ with domain $\mathcal{X}$ is said to have a Lipschitz bound $k$ for a p-norm if $\|f(x) - f(y)\|_p \leq \|x - y\|_p \forall x, y \in \mathcal{X}$. Hereafter, we assume $p=2$ so that the result is scaled with the spectral norm and omit it for simplicity. For a matrix $\boldsymbol{X}$, $\|\boldsymbol{X}\| = (\sum_{i,j} |\boldsymbol{X}_{ij}|^p)^{1/p}$ analogous to a vector norm. Before moving on, note that expansion operation can be written as below:

$$\boldsymbol{x} - \boldsymbol{C} = (\boldsymbol{x}_i - \boldsymbol{C}_{ij})_{i \in [d], j \in [h]} = \begin{pmatrix} \boldsymbol{x}_1 \mathbf{1}^T - \boldsymbol{C}_{1,:}^T \\ ... \\ \boldsymbol{x}_d \mathbf{1}^T - \boldsymbol{C}_{d,:}^T \end{pmatrix} \text{,where } \mathbf{1} \text{ is an one vector}$$

,where $\boldsymbol{x} \in R^d$ and $\boldsymbol{C} \in R^{d \times h}$. Note that expansion operation is not a linear transformation, since $(\boldsymbol{x} + \boldsymbol{y}) - \boldsymbol{C} \neq (\boldsymbol{x} - \boldsymbol{C}) + (\boldsymbol{y} - \boldsymbol{C})$.

**1**  Expansion operation and following elementwise activation function $\sigma(\cdot)$ with Lipschitz constant $\rho$, that is $\boldsymbol{x} \mapsto \sigma(\boldsymbol{x} - \boldsymbol{C})$ has a Lipschitz bound $\rho h^{1/p}$: For any two vectors $\boldsymbol{x}, \boldsymbol{y} \in R^d$, $\|\boldsymbol{x} - \boldsymbol{C} - \boldsymbol{y} + \boldsymbol{C}\| \leq \|\boldsymbol{x} - \boldsymbol{y}\| \|\mathbf{1}\| = h^{1/p} \|\boldsymbol{x} - \boldsymbol{y}\|$ by Cauchy-Schwarz inequality. By Lemma A.1, elementwise activation function at most multiplies the Lipschitz bound by $\rho$.

**2**  The Lipschitz bound of a function $\boldsymbol{x} \mapsto \boldsymbol{W} \odot \sigma(\boldsymbol{x} - \boldsymbol{C}) + B$ is $\|\boldsymbol{W}\| \rho h^{1/p}$: The proof is similar to the previous one. $\|\boldsymbol{W} \odot (\sigma(\boldsymbol{x} - \boldsymbol{C}) - \sigma(\boldsymbol{y} - \boldsymbol{C}))\| \leq \|\boldsymbol{W}\| \|\sigma(\boldsymbol{x} - \boldsymbol{C}) - \sigma(\boldsymbol{y} - \boldsymbol{C})\| \leq \|\boldsymbol{W}\| \rho h^{1/p} \|\boldsymbol{x} - \boldsymbol{y}\|$, since for any two matrices $\boldsymbol{X}, \boldsymbol{Y} \in R^{n \times m}$, $\|\boldsymbol{X}\| \|\boldsymbol{Y}\| = (\sum_{i,j} |\boldsymbol{X}_{ij}|^p)^{1/p} (\sum_{s,l} |\boldsymbol{Y}_{sl}|^p)^{1/p} = (\sum_{s=i,l=j} |\boldsymbol{X}_{ij} \boldsymbol{Y}_{sl}|^p)^{1/p} + (\sum_{s \neq i \text{ or } l \neq j} |\boldsymbol{X}_{ij} \boldsymbol{Y}_{sl}|^p)^{1/p} \geq \|\boldsymbol{X} \odot \boldsymbol{Y}\|$. Note that this corresponds to multivariate ExU.

**3**  The Lipschitz bound of DcU, $\boldsymbol{x} \mapsto \boldsymbol{A} = \sigma(\boldsymbol{W} \odot (\sigma(\boldsymbol{x} - \boldsymbol{C}^l) + \sigma(\boldsymbol{C}^r - \boldsymbol{x})) + \boldsymbol{B})$ is a $2\|\boldsymbol{W}\| \rho^2 h^{1/p}$: Compared to multivariate ExU, DcU have an additional $\boldsymbol{W} \odot \sigma(\boldsymbol{C}^r - \boldsymbol{x})$, whose Lipschitz bound is same as that of $\boldsymbol{W} \odot \sigma(\boldsymbol{x} - \boldsymbol{C}^l)$. Thus by sub-additivity matrix norm, it doubles the Lipschtiz bound. Extra activation function multiplies $rho$ to the Lipschitz bound by Lemma A.1.

**4**  $R_{max}$ keeps the Lipschitz bound: Evident by Lemma A.2, since $R_{max}$ is a pooling operation over disjoint indices of inputs.

Therefore, the Lipchitz bound of SIL is as below:

$$\|SIL(\boldsymbol{x}) - SIL(\boldsymbol{y})\| \leq 2h^{1/p} \rho^2 \|\boldsymbol{W}\| \|\boldsymbol{M}\| \|\boldsymbol{x} - \boldsymbol{y}\| \tag{4}$$

On the other hand, L-layered MLP, which is a sequence of Matrix-vector product with elementwise activation function, has the following Lipschitz bound, which is evident by Lemma A.1 and the

definition of spectral norm(Equation 3):

$$\|MLP(\boldsymbol{x}) - MLP(\boldsymbol{y})\| \leq \rho^L \prod_{l=1}^{L} \|\boldsymbol{W}_l\|_\sigma \|\boldsymbol{x} - \boldsymbol{y}\| \qquad (5)$$

.

## C  OTHER DETAILS ON SINN

Initializations of $\boldsymbol{W}, \boldsymbol{C}_l, \boldsymbol{C}_r$ are drawn from distributions whose ranges are bounded into -1 to 1, since we scale our features to -1 to 1. Specifically, TruncatedNormal for $\boldsymbol{W}$ and Uniform for $\boldsymbol{C}_l, \boldsymbol{C}_r$. Initialization of $\boldsymbol{B}$ and $\boldsymbol{M}$ are set to zeros and Xavier normal(Glorot & Bengio, 2010) respectively. It was not helpful to exponentialize weights as in the original paper or even its variants $e^x - e^{-x}$ introduced in Kim et al. (2022).

We use max as a reduction operation. This enables explainability and improves performance, compared to average. The concrete reason is not studied, but we doubt it is due to the similarity to the feature selection process, as decision tree does.

We use swish(Ramachandran et al., 2017) for activation function of MLP, not ReLU, soley because it almost always performs better. Our conjecture is that negative-allowing property of swish has a positive effect on performance, given wide input range, which contains negative values.

The baseline of Integrated Gradients should be calculated as minimum values of each units of $\boldsymbol{a}$. However due to the distribution shift, there should exist lower values. Therefore we set the baseline as $min_i \boldsymbol{a}_i - 0.1 * (max_i \boldsymbol{a}_i - min_i \boldsymbol{a}_i)$ for all $i \in [h]$. The number of steps is set to 250, while it does not much influence the result.

Batch Normalization(Ioffe & Szegedy, 2015) always drops performances both on training and test datasets. Although we did not delve into the reason, it is known that Batch normalization makes the parameter space smooth(Santurkar et al., 2018). We guess it is related to the flat minimum as we discussed.

Table 5: Candidates of Hyperparameters unspecified before : U and logU refer to Uniform and logUniform. % means either distributions are chosen at a given probability. * means 0 is sampled instead at 5% probability. ; $u$ denotes it samples by a $u$ unit

| Model | Hyperparameter | Distribution |
|-------|----------------|--------------|
| SINN  | $\beta$: Activation shape | U(1,20;0.1) |
|       | $\lambda$: Interval Norm | U(0.0,1e-3;1e-4) |
|       | $B$: Batch size | (Small) U(128,1024;32) (Large) U(512,4096;128) |
|       | $h^1$: SIL units | U(50,500;10) |
|       | $h_2$: MLP units | U(100,1000;20) |
|       | $p^1, p^2$: Dropout(SIL), Dropout(MLP) | U(0.0,0.5;0.05) |
|       | $l$: Learning rate | (50%) U(1e-4,1e-3;1e-4) (50%) U(1e-3,4e-3;5e-4) |
| XGB   | learning_rate | logU(1e-2,0.9) |
|       | max_depth | U(2,14;1) |
|       | subsample, colsample_bytree,colsample_bylevel | U(0.2,1.0) |
|       | min_child_weight | logU(1e-4,10,1e-4)* |
|       | lambda , alpha | logU(1e-5,5,1e-4)* |
|       | gamma | logU(1e-5,0.5;1e-4) |

## D  EXPERIMENT PROCEDURE

SINN is implemented using TensorFlow(Abadi et al., 2016) and our experiment requires up to 16GB VRAM. The procedure of experiments can be divided into 3 step: Preprocess, Test split and Train.

**Preprocess**    We perform minimal preprocessing. Encode categorical features into integer label, then scale each features such that their values fall between -1 to 1 in train data(minmax scale). Unseen categories in test data is assign to -2. For a regression task, we scale its label into 0 to 1.

**Test Split**    If a dataset specifies a split ratio, we use it. Otherwise, if test data is not set aside, we split the data in to 7 :1 : 2(train :validation :test). If a test set is provided, we split the remaining data into 8 : 2(train : validation). Split ratios are recorded in Table 3

**Train**    We train models with early stopping over validation data, which is not reused. For SINN, we set maximum epoch to 2000 with early stop patience being 200. Adam(Kingma & Ba, 2014) is used to train parameters. The maximum number of weak learners for XGBoost is set to 10000 with early stop patience being 300. Hyperparameters are optimized by relevant metrics for each datasets(Metric in Table 3) on validation data with HyperoptBergstra et al. (2013). Maximum trials of hyperparameter search is 100, with warmup 20 and early stop 20. During a warmup, random search is performed; afterward, hyperparameter optimization is carried out using the TPE algorithm. Hyperparameter candidates are listed in Table 5.

For SINN, Learning rate($l$) is decayed slowly multiplying 0.999 at the end of every epoch down to 10% of its start value. The MLP is fixed to two layers, whose first and second layers have $h^2$ and $h^2/2$ units respectively. The XGBoost candidate set is similar to that from Shwartz-Ziv & Armon (2022), but with some adjustments(due to infeasible ranges of hyperparameters).

SAINT is trained on the same hyperparameters as described in Borisov et al. (2022), only increasing the maximum number of epoch to 200, retrieving the best.

# E    INFORMATION ON DATASETS

Some datasets have predefined split rule, for example, Rossmann sets aside future samples(year=2015) for test data, Year sets aside last 51,630 examples to test set to avoid the 'producer effect'. Rossman and Churn are processed same as Catboost does(Rossmann-process,Churn-process). In Sarcos, torques columns are dropped, except first one, since others are basically targets. We share the urls of datasets used in this paper(hyperlinked): Adult, Blastchar, Churn, Census, CoverType, Eye, Gas, Gesture, Shrutime, Rossmann, Sarcos, Year, California, Heloc. Higgs

