# OpenReview forum: "Hidden units for tabular data representing intervals"
_ICLR.cc/2026/Conference — ICLR 2026 Conference Withdrawn Submission_

### Official Review · Reviewer_fmCy · 2025-10-27

**Soundness:** 3
**Presentation:** 2
**Contribution:** 2
**Rating:** 2
**Confidence:** 3

**Summary:**

This paper proposes Double-centered Units (DcU) and the Soft Interval Neural Network (SINN), a new neural architecture for tabular data that aims to mimic the discrete, interval-based inductive biases of tree models.

**Strengths:**

- The paper is well motivated, recognizing the inherent smoothness limitations of neural networks on tabular data and addressing this gap through interval-like representation learning.
- The generalization measurement analysis is insightful and well-executed.

**Weaknesses:**

- **Limited theoretical depth**: While the Lipschitz bound is computed, the connection between “large Lipschitz constants” and improved generalization in tabular settings remains heuristic.
There is no formal analysis explaining why non-affine transformations are fundamentally advantageous.

- **Limited comparison to recent strong neural baselines**: Only SAINT and XGBoost are included in the comparison.
It would be helpful to compare against well-known tabular neural architectures such as: FT-Transformer, TabNet... to validate generality across modern approaches.

- **Regression performance and generality**: It is unclear whether the degradation in regression performance is an inherent limitation of the proposed architecture or merely a consequence of the MSE loss and chosen optimization setup, and this distinction is crucial for evaluating the generality of the approach.
A more comprehensive study using alternative regression losses — such as Huber loss, quantile regression, or heteroscedastic objectives — would help clarify whether the observed weakness is due to the interval-based inductive bias itself, or simply a mismatch between the model and the loss function.

*Minor Issues*

- Section organization could be improved: Section 3.1 primarily reviews the ExU formulation rather than introducing the proposed method. Moving Section 3.1 into Related Work or Background would improve the overall clarity and ensure that Section 3 focuses solely on the proposed contribution (DcU and SINN).

- Writing clarity and redundancy: Some minor wording errors and repeated phrases may distract readers.

     Examples include:

     “we hypothesize that we hypothesize that…” (Section 1)

     “settle settle into minima…” (Section 2)

     A careful proofreading pass is recommended.

- Figure references and transitions: Figures are sometimes referenced without sufficient narrative setup, which makes the flow abrupt.
Adding brief context before each figure and making captions more takeaway-oriented would improve readability.
For example: “Figure X validates the effectiveness of DcU by…” instead of simply placing the figure after a sentence.

**Questions:**

See Weaknesses.

---

### Official Review · Reviewer_BPdj · 2025-10-30

**Soundness:** 2
**Presentation:** 2
**Contribution:** 2
**Rating:** 4
**Confidence:** 3

**Summary:**

This is an expert review of the research paper Hidden Units for Tabular Data Representing Intervals.

Detailed Summary
The research addresses the fundamental challenge that tree-based boosting models maintain dominance over standard neural networks for tabular data. This is attributed to the fact that conventional neural units impose overly smooth inductive biases which are unsuitable for the distinct representations required by this data type. The primary purpose of this work is to introduce novel hidden units and a corresponding supervised neural network architecture that can effectively learn tabular representations by incorporating rapid output changes and sudden gradient shifts.

The limitations of preceding methods are centered on excessive smoothness. Specifically, the prior Exponential-centered Units or ExU units suffer from a hindered training process where a sharp slope results in gradients vanishing across almost all points, stalling the unit parameter learning. The poor utility of ExU was evidenced by the large number of units, one thousand, required to fit simple Bernoulli data.

The core idea is the introduction of the Double-centered Unit or DcU. This unit is an improvement over ExU as it parameterizes soft intervals using two center parameters, which enables it to activate and preserve informative gradients outside the learned interval. This design successfully resolves the gradient vanishing issue. Building on this unit, the proposed model is the Soft Interval Neural Network or SINN, an encoder-Multilayer Perceptron architecture that employs a DcU-based Soft Interval Layer with max pooling and interval sparsity regularization. The author posits that the non-affine operations within the Soft Interval Layer allow SINN to expand its L2 margin even in low-dimensional spaces, a necessary robustness advantage for tabular data where key subgroups may be defined by only a few features.

The performance of the proposed method is verified through extensive supervised learning experiments across fifteen public datasets. The model is benchmarked against established methods like XGBoost and SAINT. Furthermore, the paper investigates common generalization proxies such as spectral or Lipschitz bounds, Hessian-based flatness, and dropout-based ensembling. This investigation empirically finds that smoothness-oriented regularization is not consistently predictive of performance in the tabular domain.

The research concludes that non-affine, interval-like representations provide a highly useful inductive bias for tabular classification problems. The main contribution of this paper is the introduction of DcU and the resulting SINN architecture, which is shown to deliver classification accuracy that is competitive with or superior to XGBoost. A significant secondary contribution is the empirical evidence that smoothness-oriented regularization is not reliably predictive of tabular performance, motivating a need for new theoretical analyses that move beyond affine assumptions.

**Strengths:**

The proposed Double-centered Units and the Soft Interval Neural Network directly address the core limitation of standard neural networks by deliberately introducing a non-affine, interval-like inductive bias. This mimics the successful split-based mechanism found in decision trees which is well-suited for tabular feature segmentation. The DcU design specifically corrects the critical gradient vanishing flaw of its predecessor unit ExU by ensuring informative gradients are preserved outside the learned soft interval, thus guaranteeing better learnability.

The primary contribution is the creation of the end-to-end neural network architecture SINN which is demonstrated to be a viable and powerful alternative to the XGBoost benchmark for tabular classification tasks. This finding validates that interval-based hidden units are a potent and necessary tool in the neural network toolkit for this data domain.

The method is logically concrete because the non-smooth operation of the Soft Interval Layer is explicitly connected to the objective of maximizing the L2 margin within low-dimensional feature spaces. This mechanism provides a principled way to overcome the inherent implicit bias limitations observed in standard Multilayer Perceptrons when applied to tabular data.

**Weaknesses:**

The paper provides only an empirical justification for selecting the softsign activation function for DcU over other symmetric alternatives. A robust theoretical or logical explanation as to why softsign is uniquely capable of maintaining non-vanishing gradients outside the soft interval is not presented. Furthermore, the observed performance gap between classification and regression is merely hypothesized to be a reflection of the implicit bias of neural networks without a deep logical analysis to back this claim.

While the classification performance is strong, the mixed results on regression tasks suggest that the proposed method does not universally solve the problem of surpassing tree-based models across all tabular learning objectives. This partial success indicates a potential limitation in the robustness or general applicability of the interval-based inductive bias.

A practical concern arises from the need for manual feature discretization on the Adult dataset to achieve competitive performance on its multiscale features. This preprocessing step suggests a limitation in the DcU's inherent ability to handle features with widely varying scales, a common characteristic of real-world tabular data. Additionally, a key finding of the paper is the failure of smoothness-oriented regularization to predict tabular performance; yet, the theoretical analysis motivated by this finding is absent, leaving the conclusion incomplete.

**Questions:**

The performance discrepancy between classification and regression is hypothesized to reflect the implicit bias of neural networks. What is the precise nature of this implicit bias, and why does the non-affine DcU bias fail to fully mitigate it for regression objectives?

Given the requirement for manual feature discretization to handle the multiscale feature of the Adult dataset, is there a planned design update or modification to the Soft Interval Layer or the Double-centered Unit to inherently resolve this limitation, or will this always remain a necessary preprocessing step for multiscale features?

The paper motivates future theoretical analysis beyond affine assumptions due to the empirical failure of smoothness-oriented generalization proxies. What is the author's anticipated path forward for this theoretical work, and what non-affine generalizations of existing generalization bounds are currently being investigated?

Can the author provide a detailed logical justification for why softsign is superior to other symmetric functions like Tanh or Sigmoid in maintaining non-vanishing gradients outside the soft interval, moving beyond the current empirical observation?

---

### Official Review · Reviewer_NPmk · 2025-10-31

**Soundness:** 3
**Presentation:** 2
**Contribution:** 2
**Rating:** 2
**Confidence:** 5

**Summary:**

This paper attempts to address the long-standing problem of deep neural networks lagging behind tree models (such as XGBoost) on tasks involving tabular data. The core hypothesis is that the oversmoothing inductive bias introduced by hidden units in existing neural networks (such as ReLU) prevents the model from learning the discrete nature of tabular data. The authors propose a new type of non-affine hidden unit that is more adaptable to tabular data, improving model performance by breaking down the oversmoothing.

**Strengths:**

S1. The paper explains the weaknesses of neural networks on tabular data from the perspective of "hidden unit inductive bias" and proposes DCU as a new unit design. This novel perspective may inspire further research on unit structure in the field of tabular learning.

S2. The paper explains the difference between ExU and DCU using the Lipschitz upper bound and gradient dynamics analysis. Furthermore, it analyzes the implicit bias of neural networks on tabular data based on assumptions, providing a useful theoretical foundation for subsequent improvements.

**Weaknesses:**

W1. The current experiments only compared two baselines, XGBoost and SAINT. Strong models representative of the Tabular benchmark, such as LightGBM[1], CatBoost[2], and TabPFN[3], were not included. Furthermore, SINN's advantages were not significant in tabular classification tasks. In most cases, it was only slightly better than XGBoost, and even performed worse on datasets like Adult. Its disadvantage was even more pronounced in regression tasks.

W2. The paper's motivation is that "oversmoothing hidden units affects model performance," which is highly relevant to domain generalization and adaptability to different data distributions. However, the current experiments are all conducted in a single IID scenario and do not analyze the effects of inter-domain transfer. Furthermore, the discussion of implicit bias only attempts to compare LB with SB, but does not include more general flatness optimization methods (such as SAM[4]).

Overall, the approach is novel, but the experiments are insufficient and the performance is not competitive.

**Reference**

[1] Ke, Guolin, et al. "Lightgbm: A highly efficient gradient boosting decision tree." Advances in neural information processing systems 30 (2017).

[2] Prokhorenkova, Liudmila, et al. "CatBoost: unbiased boosting with categorical features." Advances in neural information processing systems 31 (2018).

[3] Hollmann, Noah, et al. "Accurate predictions on small data with a tabular foundation model." Nature 637.8045 (2025): 319-326.

[4] Foret, Pierre, et al. "Sharpness-aware minimization for efficiently improving generalization." arXiv preprint arXiv:2010.01412 (2020).

**Questions:**

Q1. Why are only XGBoost and SAINT selected as the baseline, and not LightGBM, CatBoost, or even TabPFN? Are there performance or computational considerations?

Q2. Beyond the LB/SB comparison experiments, have you tested the impact of SAM or other flatness optimization algorithms on SINNs?

Q3. Does the interval parameter of DCU exhibit clear feature selection behavior during training?

---

### Official Review · Reviewer_Z37y · 2025-11-02

**Soundness:** 1
**Presentation:** 2
**Contribution:** 1
**Rating:** 2
**Confidence:** 4

**Summary:**

The paper argues that standard MLPs have the wrong inductive bias for tabular data because they are too smooth, while tabular patterns are often threshold/interval-like. It introduces a new hidden unit, the Double-centered Unit (DcU), which explicitly learns soft intervals via two learnable boundaries and keeps gradients nonzero outside the interval, fixing training/initialization issues seen in earlier ExU-style units. Building on this, the authors propose SINN (Soft Interval Neural Network): a front “soft-interval layer” that maps each feature into multiple learnable intervals, then aggregates them (with max) and feeds a small MLP, giving both good accuracy and straightforward feature attribution. Across 15 tabular datasets, SINN matches or beats strong neural baselines and is often competitive with XGBoost on classification, while experiments show that “making the model smoother” (spectral norm, flat-minima tricks, strong dropout) does not reliably improve tabular performance—supporting the claim that tabular models should favor non-affine, interval-like representations.

**Strengths:**

1. Well-motivated problem: clearly identifies a real gap between standard MLP inductive bias (smooth, affine-stacked) and the threshold/interval structure common in tabular data, and argues for a different primitive.

2. New hidden unit (DcU): improves over ExU by using two learnable boundaries and keeping gradients outside the interval, which is a simple but nontrivial architectural contribution.

3. Empirical competitiveness on classification: across a diverse set of tabular benchmarks, the method is often on par with or better than strong neural baselines and close to XGBoost

**Weaknesses:**

1. The paper claims that standard MLPs have an inappropriate inductive bias for tabular data because they are overly smooth, which in turn leads to suboptimal performance on tabular learning. However, well-designed and well-tuned MLPs can in fact achieve very strong results. For example, RealMLP [1] performs quite well on the Talent [2] and TabArena [3] benchmarks, which runs counter to the authors’ central argument.

2. Using only 15 tabular datasets is insufficient to draw conclusions that are broadly convincing.

3. Computational/practical overhead: SINN is implemented in TensorFlow, and the authors report needing up to 16GB of GPU VRAM to run their experiments, which is noticeably heavier than running XGBoost or SAINT on the same tabular tasks.

4. Regression performance is weak, even though the authors recognize this limitation and provide some explanations.

[1] Holzmüller, David, Léo Grinsztajn, and Ingo Steinwart. "Better by default: Strong pre-tuned mlps and boosted trees on tabular data." Advances in Neural Information Processing Systems 37 (2024): 26577-26658.
[2] Ye, Han-Jia, et al. "A closer look at deep learning methods on tabular datasets." arXiv preprint arXiv:2407.00956 (2024).
[3] Erickson, Nick, et al. "Tabarena: A living benchmark for machine learning on tabular data." arXiv preprint arXiv:2506.16791 (2025).

**Questions:**

1. Your main claim is that standard MLPs are too smooth for tabular data, but strong, well-tuned MLPs (e.g. RealMLP) work very well on recent tabular benchmarks. How do you explain this?

2. Please provide a clearer computational cost analysis: VRAM needed per dataset/model, training time, and comparison to XGBoost

3. The evaluation uses only 15 datasets. Can you add more tabular datasets to make the conclusions more convincing?

---

### Note · Authors · 2025-11-19

**Comment:**

Thank for your sincere comments. Judging upon the comments we received, our submission is not good enough and we decide to withdraw our submission. Hope ICLR 2026 be held and finish successfully.

**Withdrawal Confirmation:**

I have read and agree with the venue's withdrawal policy on behalf of myself and my co-authors.